# Spatiotemporal and kinetic characteristics during maximal sprint running in fast running soccer players

Yohei Takai[ID][1]*, Terumitsu Miyazaki[1], Norihide Sugisaki[2], Takaya Yoshimoto[3], Naotoshi Mitsukawa[4], Kai Kobayashi[5], Hiroyasu Tsuchie[6], Hiroaki Kanehisa[1]

**1** National Institute of Fitness and Sports in Kanoya, **2** Center for Liberal Arts, Meiji Gakuin University, **3** Faculty of Welfare Society, The International University of Kagoshima, **4** Faculty of Human Sciences, Toyo Gakuen University, **5** Faculty of Information Sciences and Arts, Toyo University, **6** Faculty of Law, Toyo University

☯ These authors contributed equally to this work.

* y-takai@nifs-k.ac.jp

## Abstract

This study aimed to elucidate spatiotemporal and kinetic variables in fast-running soccer players in comparison with sprinters or slow-running soccer players. Sixty-seven male soccer players and 17 male sprinters ($Sp$) performed 60-m maximal effort sprint running. The soccer players were classified into three groups: high-speed ($SOC_{High}$), medium-speed, and low-speed ($SOC_{Low}$). The antero-posterior and vertical ground reaction forces were measured with a 50-m long force plates system at every step during the sprint. Step length and step frequency were also computed from the position of center of pressure, contact time, and flight time. During the initial acceleration phase, $SOC_{High}$ exhibited similar running speeds to $Sp$. This was attributed to a higher step frequency in $SOC_{High}$ compared to $Sp$, while net antero-posterior impulse was lower in the former than in the later. In the range of running speed from 7.5 m/s to 8.5 m/s, net antero-posterior impulse for $SOC_{High}$ was similar to that for $Sp$. At 9.0 m/s, $SOC_{High}$ exhibited a lower net antero-posterior impulse compared to $Sp$, primarily due to a reduced propulsive impulse. Additionally, vertical impulse during the braking phase was larger in $SOC_{High}$ compared to $Sp$, due to a longer braking time, while vertical impulse during the propulsive phase was smaller, due to a tendency for a reduced propulsive time and vertical force during the corresponding phase. Compared to $SOC_{Low}$, $SOC_{High}$ exhibited higher step frequency through sprint running and longer step lengths from the 2nd acceleration phase to maximal speed phase. Additionally, net antero-posterior impulse at the same running speed was greater in $SOC_{High}$ compared to $SOC_{Low}$. Vertical impulse was lower during the braking phase but higher during the propulsive phase in $SOC_{High}$ than in $SOC_{Low}$. Thus, the sprint mechanics of $SOC_{High}$ is characterized by a similar ability of speed acquisition up to the 2nd acceleration as sprinters. However, at 9.0 m/s or over, $SOC_{High}$ exhibits a greater vertical impulse, leading to a lower step frequency.

**Data availability statement:** All relevant data are within the paper and its Supporting Information files.

**Funding:** The author(s) received no specific funding for this work.

**Competing interests:** The authors have declared that no competing interests exist.

## Introduction

In soccer players, sprint running ability is one of the crucial factors related to competition performance [1, 2]. Approximately 10% of the total distance covered during a game (ranging from 10–12 km) involves near-straight running [3]. Moreover, high-speed straight running often precedes scoring opportunities for both offensive players and defenders [1,2,4]. The relative distribution of high-speed running (defined as speeds >6.67 m/s) is approximately 7% for distances of ~10m, 48% for distances of ~20m, and 45% for distances exceeding 20m [5]. This implies that soccer players are required to run over both short and long distances at high speeds during a game. In essence, both high acceleration ability and maximum sprint speed are crucial for high level soccer players. Therefore, elucidating the mechanics of maximal sprint running in fast-running soccer players during the acceleration and maximal speed phases will provide useful information for designing effective training programs in practical fields.

There are three primary phases during sprint running to achieve maximal speed: the 1st acceleration, the 2nd acceleration, and the maximal speed phases [6–9]. Although maximal running speed depends on speed gain in the preceding acceleration phase [10], the correlation coefficients between acceleration and maximal running speeds have been shown to range from 0.56–0.79 [11–13], indicating potential differences in sprint mechanics between the acceleration and maximal running speed phases [14]. Thus, examining sprint mechanics during both phases will provide useful information to elucidate the mechanics for achieving high sprint running performance.

Spatiotemporal and kinetics parameters of sprint running during acceleration and maximal speed phases have been observed to differ between competitive sprinters (hereafter referred to as sprinters) and athlete non-sprinters (non-sprinters), as well as among non-sprinters with varying running capabilities within the same event [15–17]. For example, it has been reported that the antero-posterior ground reaction forces (GRFs) varied between faster and slower sprinters throughout the acceleration phase [18]. Further, the propulsive and braking forces were lower in soccer players than in sprinters at equivalent speed (8 and 8.5 m/s) during the 2nd acceleration phase [15]. During the maximal speed phase, non-sprinters exhibit longer contact times and shorter step lengths compared with sprinters [19]. As compared to non-sprinters, sprinters demonstrate larger vertical GRFs and exert greater vertical GRFs during the first half of ground contact in the maximal speed phase [16]. Furthermore, elite soccer players demonstrate a higher theoretical maximal horizontal force derived from horizontal force-velocity relationship compared with sub-elite soccer players [17], indicating that the formers have greater acceleration capability in the initial acceleration phase. Previous studies have shown that soccer players exhibit inferior horizontal force exertion at high running speeds and differ in the spatiotemporal variables during maximal speed phase compared with sprinters [15]. Additionally, high-level soccer players exhibit superior antero-posterior force production in the acceleration phase compared with low-level players [17]. However, current literature does not clarify whether fast-running soccer players can

generate antero-posterior GRFs similar to that of sprinters in the initial acceleration phase. It remains unknown how the vertical components of GRFs differ among soccer players with different running speeds, as well as between sprinters and fast-running soccer players in the 2nd acceleration and maximal speed phases.

Therefore, this study aimed to elucidate the differences in spatiotemporal and kinetics parameters between sprinters and fast-running soccer players, and between fast- and slow-running soccer players. We hypothesized that: 1) in the 1st acceleration phase, antero-posterior force production and spatiotemporal variables for fast-running soccer players would be similar to that of sprinters and superior to that of slow-running soccer players; 2) in the 2nd acceleration and the maximal speed phases, fast-running soccer players would produce lower vertical force and would have lower step frequency and step length compared with sprinters.

## Methods

### Participants

Data were collected from December 21, 2021 to September 17, 2023. Sixty-seven male soccer players (*SOC*) and 17 male sprinters (*Sp*; 20.6 ± 0.9 years, 1.73 ± 0.05 m, 67.7 ± 5.1 kg; mean ± SD) participated in this study. All participants underwent their specific training 5-6 times per week, with each session lasting 1.5-2 hours. None of the participants reported any illness and were not prescribed medications for cardiovascular, metabolic, or orthopedic disorders. According to the athlete taxonomy classification reported [18], the study participants categorized as Tier 3 or Tier 4. The participants had been competing in sprint events for more than 5 years. Based on the maximal running speed (as described in the data analysis section), soccer players were divided into three groups: high-speed running soccer players whose speed was above the mean + 0.5SD ($SOC_{High}$; n = 16, 20.1 ± 0.9 years, 1.72 ± 0.05 m, 68.1 ± 7.3 kg), medium-speed soccer players whose speed was within the mean ± 0.5SD ($SOC_{Med}$; n = 30, 20.3 ± 1.1 years, 1.73 ± 0.05 m, 67.5 ± 5.4 kg), and low-speed soccer players whose speed was below the mean - 0.5SD ($SOC_{Low}$; n = 21, 20.2 ± 1.9 years, 1.71 ± 0.05 m, 65.7 ± 5.3 kg). Physical characteristics of the participants did not significantly differ between groups. This study was approved by the ethics committee of the National Institute of Fitness and Sports in Kanoya (#5-60). All procedures were conducted in accordance with the Declaration of Helsinki. Prior to the experiments, all the participants were fully informed of the purpose and risks of the experiments and written consent was obtained.

### Experimental protocol

Prior to the test session, all participants engaged in approximately 30 min of warm-up exercises, comprising of approximately 8 min of jogging to running, approximately 10 min of static and dynamic stretching, approximately 10 min of running technique exercises, 2–4 submaximal-to-maximal sprint running. Following at least 10 min of rest after the warm-up session, the participants performed maximal effort sprint running from standing split-stance position on an indoor track with 54 force plates (TF-90100, TF-3055, and TF-32120; Tec Gihan, Uji, Japan), as described in previous studies [7,19]. The soccer players wore their own running shoes, and the sprinters used their own spikes.

### Data analysis

Antero-posterior and vertical force signals were used for analysis. The analog force signals from each force plate were captured in a control box (FP Control Unit, Tec Gihan, Japan), converted from analog to digital, and transmitted to a computer. The 50-m long force plates were treated as a single unit using dedicated software, and the antero-posterior and vertical forces and the location of the center of pressure on the track during ground contact, were computed at a sampling rate of 1000 Hz. Fig 1 presents representative data on antero-posterior and vertical ground reaction forces for $SOC_{High}$, $SOC_{Low}$, and *Sp*.

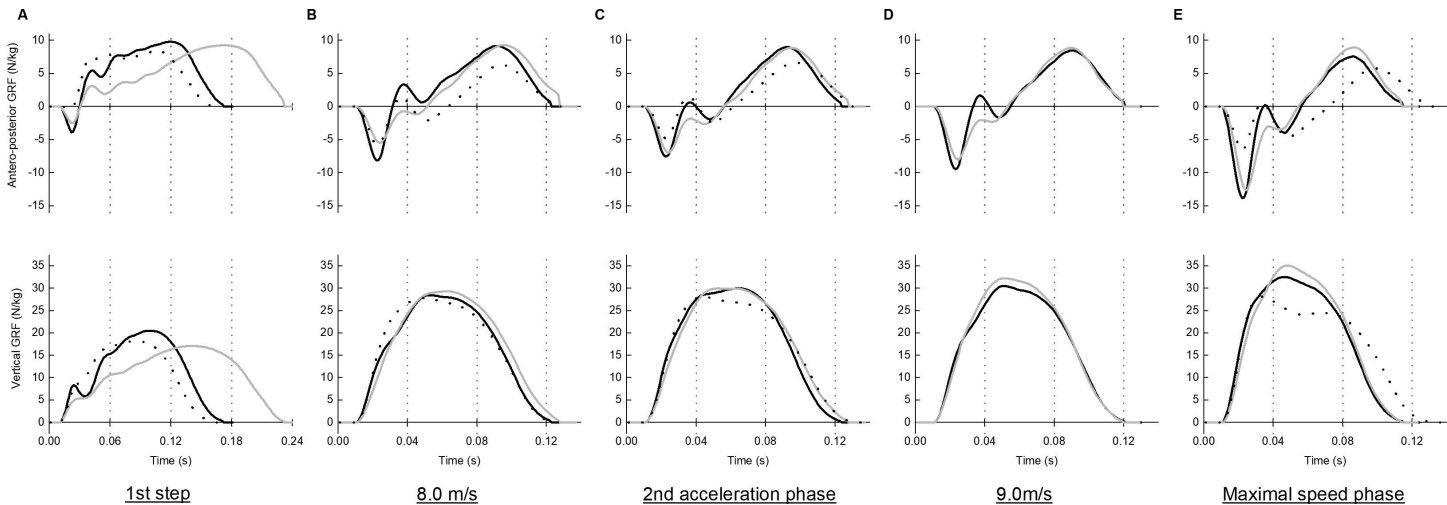

**Fig 1. Representative data on antero-posterior and vertical ground reaction forces (GRFs) for a sprinter, a fast-running soccer player, and a slow-running soccer player across the analyzed steps.** (A) First step, (B) Running speed of approximately 8.0 m/s, (C) Second acceleration phase, (D) Running speed of approximately 9.0 m/s, and (E) Maximal speed phase. The solid black line represents a fast-running soccer player, the solid gray line represents a sprinter, the dotted black line represents a slow-running soccer player.

In accordance with the methodology described in our previous study [19], the force data underwent filtering using a fourth-order zero-lag low-pass Butterworth filter with a cutoff frequency of 50 Hz. Ground contact and takeoff times were determined based on thresholds set at 20 N for vertical force, allowing identification of the braking and propulsion phases based on negative and positive values of the antero-posterior force, respectively. Spatiotemporal parameters were calculated as described in previous studies [19,20]. Ground contact time was defined as the duration between ground contact and takeoff for each step. Flight time was the period between takeoff in one step and ground contact in the next step. Step frequency was calculated as the inverse of the sum of contact time and flight time, and step length was determined as the distance between the midpoint of ground contact and takeoff in one step and the corresponding midpoint in the next step. Running speed was obtained by multiplying step frequency and step length. Antero-posterior and vertical forces were averaged over the duration from ground contact to takeoff for each step, and net antero-posterior and vertical impulses were computed through time integration of these forces. To mitigate the influence of left-right imbalance on the measured variables, the step-by-step data were smoothed using a moving average over two steps. In accordance with a previously published study [21], the average value of each variable was calculated for each participant during the first four steps ($A_{1st}$), the middle four steps (specific to individuals) ($A_{2nd}$), and four steps including maximal speed (including two steps before and one step after maximal speed ($MS$). The middle four steps were defined relative to the middle step during the acceleration phase from the start to the step at maximal speed.

To examine group differences in each variable at the same running speed, we interpolated each variable as a function of running speed within the data range of each participant. From the interpolated data, we extracted the measured variables at 7.5 m/s, 8.0 m/s, 8.5 m/s, and 9.0 m/s. In $SOC_{Low}$, the running speed of two participants did not reach a maximum of 8.0 m/s. In $SOC_{Med}$, three participants did not reach 8.5 m/s, while in $SOC_{High}$, two participants did not reach 9.0 m/s. In $Sp$, one participant also did not reach 9.0 m/s. Therefore, the independent variables at the corresponding running speed were treated as missing values. To determine the theoretical maximal running speed ($MS_0$), the stance-averaged horizontal forces were plotted against running speed and fitted linearly with a first-order polynomial function. The y-intercept was computed as $MS_0$. The running speed in each phase was also expressed as the value relative to $MS_0$ (%$MS_0$).

## Statistical analysis

Descriptive data are presented as means and standard deviations. Given the assumption of group differences in running speed, a generalized linear mixed effect model (GLMM) was employed to examine group differences in the independent variables. The model incorporated group ($Sp$, $SOC_{High}$, $SOC_{Med}$, and $SOC_{Low}$), phase (7.5 m/s, 8.0 m/s, 8.5 m/s, 9.0 m/s, $A_{1st}$, $A_{2nd}$, and $V_{max}$), and a group-by-phase interaction. The GLMM included a random intercept to accommodate random variation in running speed at the maximal speed phase between participants and a random slope for phase. This was done to address random variation changes in the independent variables across phases between the participants. Covariation between the random intercept and slope was assessed to determine if the change across phases was dependent on running speed in the maximal speed phase. When significant main effects of group and phase were detected, a Bonferroni test was conducted with all combinations for post hoc comparisons. Additionally, in the presence of a significant group-by-phase interaction, a GLMM with a Bonferroni test was applied within each group to assess the significance of phase-related differences in the independent variables. One-way analysis of variance (ANOVA) with factorial analysis was conducted to test group differences in mechanical parameters derived from the stance-averaged GRFs and running speed relationship. In this analysis, when the $F$ value was significant, a Bonferroni test was conducted for post hoc comparisons. The level of significance was set at 5%, and the $p$-value was computed using statistical software (IBM SPSS Statistics 26, IBM, Japan).

## Results

Fig 2 summarizes the main results of this study. To characterize the spatiotemporal and kinetic parameters of fast-running soccer players, we compared each variable between $SOC_{High}$ and $Sp$ and between $SOC_{High}$ and $SOC_{Low}$.

### Initial sprint acceleration phase ($A_{1st}$)

Table 1 presents running speed and spatiotemporal variables during sprint running for all groups. No significant difference was found in running speed between $SOC_{High}$ and $Sp$, but the $\%MS_0$ tended to be higher in $SOC_{High}$ compared with $Sp$ ($p$ = 0.061). Running speed was significantly higher in $SOC_{High}$ than in $SOC_{Low}$. $SOC_{High}$ exhibited a shorter propulsive time, resulting in a shorter ground contact time and a higher step frequency compared to $Sp$ and $SOC_{Low}$ (Table 1). There were no significant group differences in the braking time. Flight time tended to be longer in $SOC_{High}$ compared to $SOC_{Low}$ ($p$ = 0.057). There was no significant difference in step length between $SOC_{High}$ and $Sp$ or between $SOC_{High}$ and $SOC_{Low}$.

Table 2 presents antero-posterior and vertical force production during sprint running. $SOC_{High}$ exhibited lower net antero-posterior impulse and propulsive impulse compared to $Sp$. No significant differences in stance-averaged antero-posterior force, braking or propulsive force were observed between $SOC_{High}$ and $Sp$. Net antero-posterior impulse was greater in $SOC_{High}$ than in $SOC_{Low}$, resulting from a greater propulsive impulse. $SOC_{High}$ exhibited a greater propulsive force compared to $SOC_{Low}$, resulting in a greater stance-averaged antero-posterior force.

The vertical impulse was lower in $SOC_{High}$ than in $Sp$ and $SOC_{Low}$ (Table 2). The averaged vertical force during the propulsive phase tended to be greater in $SOC_{High}$ compared with $SOC_{Low}$ ($p$ = 0.058).

### The second acceleration phase ($A_{2nd}$)

Running speed was lower in $SOC_{High}$ than that in $Sp$ but higher than in $SOC_{Low}$ (Table 1). There were no significant group differences in the $\%MS_0$. No significant difference in ground contact time was found between $SOC_{High}$ and $Sp$ (Table 1). $SOC_{High}$ exhibited a shorter ground contact time compared to $SOC_{Low}$, primarily due to a shorter propulsive time. There was no significant difference in flight time between $SOC_{High}$ and $Sp$, nor between $SOC_{High}$ and $SOC_{Low}$. Step length was shorter in $SOC_{High}$ compared to $Sp$, while no significant difference in step frequency was observed between the two group. $SOC_{High}$ demonstrated a higher step frequency and a longer step length compared to $SOC_{Low}$.

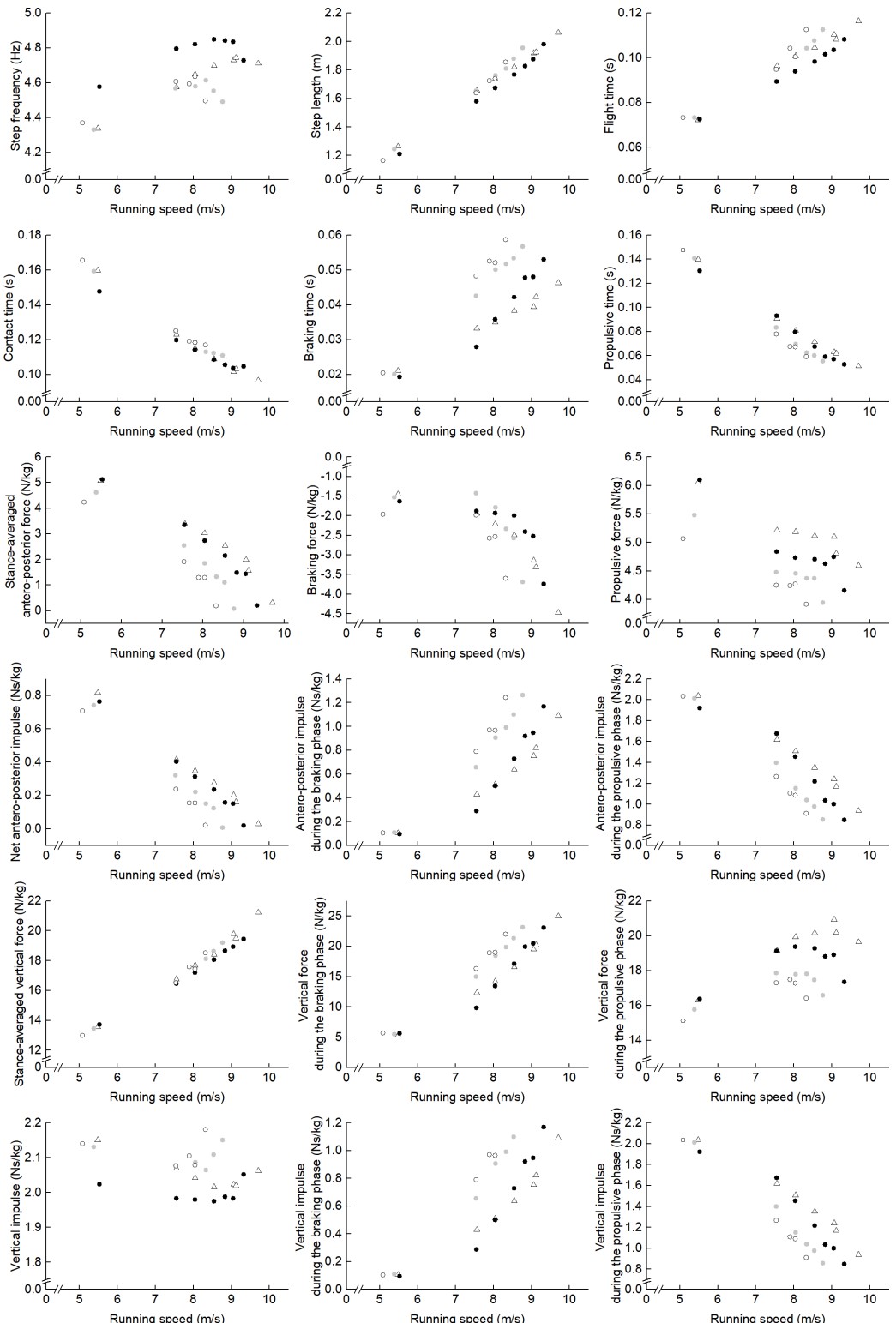

**Fig 2. Associations of spatiotemporal variables and ground reaction forces with running speed.** Data points are represented as sprinters (white-filled triangles), fast-running soccer players (black-filled circles), medium-running soccer players (gray-filled circles), and slow-running soccer players (white-filled circles).

**Table 1. Group differences in number of steps, analytical distance from start line, and spatiotemporal parameters during sprint running.**

| | | Sprinters | High-speed soccer players | Medium-speed soccer players | Low-speed soccer players | Group difference | Within-subject effect (p-value) |
|---|---|---|---|---|---|---|---|
| # of steps | $A_{1st}$ | 13 ± 1 | 14 ± 1 | 14 ± 1 | 15 ± 1 | | |
| | $A_{2nd}$ | 26 ± 1 | 28 ± 2 | 28 ± 1 | 29 ± 1 | | |
| | MS | — | — | — | — | | |
| Distance from start line, m | $A_{1st}$ | 2.2 ± 0.7 * | 1.5 ± 0.2 * | 1.7 ± 0.2 * | 1.5 ± 0.2 * | | Gp 0.001 |
| | $A_{2nd}$ | 20.0 ± 1.4 # | 18.6 ± 2.7 # | 17.5 ± 3.1 # | 16.1 ± 3.6 # | Sp>L | Ph <0.001 |
| | MS | 42.5 ± 1.5 | 40.1 ± 3.9 | 38.7 ± 5.5 | 36.1 ± 6.5 | Sp>M,L, H>L | Gp×Ph <0.001 |
| Running speed, m/s | $A_{1st}$ | 5.49 ± 0.18 * | 5.53 ± 0.24 * | 5.38 ± 0.15 * | 5.08 ± 0.21 * | Sp,H,M>L | Gp <0.001 |
| | $A_{2nd}$ | 9.12 ± 0.40 # | 8.83 ± 0.29 # | 8.33 ± 0.22 # | 7.90 ± 0.31 # | Sp>Soc, H>M>L | Ph <0.001 |
| | MS | 9.71 ± 0.44 | 9.33 ± 0.27 | 8.77 ± 0.13 | 8.32 ± 0.24 | Sp>Soc, H>M>L | Gp×Ph <0.001 |
| $\%MS_0$, % | $A_{1st}$ | 52.8 ± 2.0 * | 55.8 ± 3.2 * | 58.0 ± 2.7 * | 57.1 ± 3.6 * | Sp<M,L | Gp 0.015 |
| | $A_{2nd}$ | 87.7 ± 1.9 # | 89.2 ± 3.9 # | 89.8 ± 4.2 # | 88.6 ± 4.2 # | | Ph <0.001 |
| | MS | 93.3 ± 1.5 | 94.2 ± 2.8 | 94.5 ± 2.6 | 93.4 ± 2.7 | | Gp×Ph <0.001 |
| Ground contact time, s | $A_{1st}$ | 0.160 ± 0.014 * | 0.148 ± 0.009 * | 0.159 ± 0.015 * | 0.165 ± 0.011 * | H<Sp,M,L, M<L | Gp <0.001 |
| | $A_{2nd}$ | 0.103 ± 0.008 # | 0.106 ± 0.007 # | 0.113 ± 0.007 | 0.119 ± 0.008 | Sp,H<M,L | Ph <0.001 |
| | MS | 0.097 ± 0.008 | 0.104 ± 0.006 | 0.111 ± 0.006 | 0.117 ± 0.006 | Sp<SOC, H<L | Gp×Ph <0.001 |
| Braking time, s | $A_{1st}$ | 0.021 ± 0.003 * | 0.019 ± 0.002 * | 0.020 ± 0.005 * | 0.020 ± 0.002 * | | Gp <0.001 |
| | $A_{2nd}$ | 0.042 ± 0.008 # | 0.048 ± 0.004 # | 0.052 ± 0.005 # | 0.053 ± 0.008 | Sp<M,L | Ph <0.001 |
| | MS | 0.046 ± 0.005 | 0.053 ± 0.004 | 0.057 ± 0.004 | 0.059 ± 0.004 | Sp<SOC | Gp×Ph <0.001 |
| Propulsive time, s | $A_{1st}$ | 0.140 ± 0.012 * | 0.130 ± 0.009 * | 0.141 ± 0.013 * | 0.147 ± 0.011 * | H<Sp,M,L, Sp<L | Gp 0.023 |
| | $A_{2nd}$ | 0.062 ± 0.006 # | 0.059 ± 0.005 # | 0.063 ± 0.006 # | 0.067 ± 0.007 | H<L | Ph <0.001 |
| | MS | 0.051 ± 0.004 | 0.053 ± 0.003 | 0.055 ± 0.005 | 0.059 ± 0.003 | Sp<L | Gp×Ph <0.001 |
| Flight time, s | $A_{1st}$ | 0.072 ± 0.012 * | 0.073 ± 0.011 * | 0.073 ± 0.011 * | 0.065 ± 0.014 * | M>L | Gp 0.020 |
| | $A_{2nd}$ | 0.108 ± 0.006 # | 0.101 ± 0.006 # | 0.104 ± 0.007 # | 0.099 ± 0.010 | Sp>L | Ph <0.001 |
| | MS | 0.116 ± 0.007 | 0.108 ± 0.007 | 0.112 ± 0.009 | 0.106 ± 0.008 | Sp>H,L | Gp×Ph <0.001 |
| Step frequency, steps/s | $A_{1st}$ | 4.34 ± 0.21 * | 4.58 ± 0.24 * | 4.33 ± 0.24 * | 4.37 ± 0.23 * | H>Sp,M,L | Gp <0.001 |
| | $A_{2nd}$ | 4.74 ± 0.20 # | 4.84 ± 0.22 # | 4.61 ± 0.16 # | 4.59 ± 0.23 # | H>M,L | Ph <0.001 |
| | MS | 4.71 ± 0.21 | 4.73 ± 0.21 | 4.49 ± 0.17 | 4.49 ± 0.20 | Sp,H>M,L | Gp×Ph 0.001 |
| Step length, m | $A_{1st}$ | 1.26 ± 0.07 * | 1.21 ± 0.10 * | 1.25 ± 0.08 * | 1.16 ± 0.08 * | Sp,M>L | Gp <0.001 |
| | $A_{2nd}$ | 1.92 ± 0.09 # | 1.83 ± 0.09 # | 1.81 ± 0.07 # | 1.72 ± 0.10 # | Sp>SOC, H>L | Ph <0.001 |
| | MS | 2.06 ± 0.08 | 1.98 ± 0.11 | 1.96 ± 0.07 | 1.86 ± 0.08 | Sp>SOC, H>L | Gp×Ph <0.001 |

Values are means and SDs. $A_{1st}$, the analytical steps across the 1st to 4th steps;

$A_{2nd}$, the analytical steps across 4 steps corresponding to half the number of steps required to reach maximal speed phase; MS, the analytical steps across 4 steps to reach maximal speed

Gp, group; Ph, phase; Gp×Ph, group by phase interaction; Sp, sprinters; Soc, soccer players; H, the high-speed soccer players; M, the medium-speed soccer players; L, the low-speed soccer players

*, significant difference compared to the other phases; $, significant difference compared to the $V_{Max}$

*, significant difference compared to the other phases; #, significant difference compared to the $A_{2nd}$;

**Table 2. Group differences in horizontal and vertical force production during sprint running.**

| Variable | Step | Sprinters | High-speed soccer players | Medium-speed soccer players | Low-speed soccer players | Group difference | Within-subject effect | (p-value) |
|---|---|---|---|---|---|---|---|---|
| Net antero-posterior impulse (Ns/kg) | $A_{1st}$ | 0.82 ± 0.05 * | 0.76 ± 0.05 * | 0.74 ± 0.05 * | 0.71 ± 0.05 * | Sp>Soc, H>L | Gp | <0.001 |
| | $A_{2nd}$ | 0.16 ± 0.02 # | 0.16 ± 0.04 # | 0.15 ± 0.05 # | 0.15 ± 0.04 # | | Ph | <0.001 |
| | MS | 0.03 ± 0.01 | 0.02 ± 0.03 | 0.01 ± 0.03 | 0.02 ± 0.03 | | Gp×Ph | <0.001 |
| Braking impulse (Ns/kg) | $A_{1st}$ | -0.03 ± 0.02 * | -0.03 ± 0.02 * | -0.03 ± 0.02 * | -0.04 ± 0.01 * | | Gp | <0.001 |
| | $A_{2nd}$ | -0.13 ± 0.01 # | -0.11 ± 0.02 # | -0.12 ± 0.03 # | -0.13 ± 0.03 # | | Ph | <0.001 |
| | MS | -0.20 ± 0.02 | -0.20 ± 0.02 | -0.21 ± 0.03 | -0.21 ± 0.03 | | Gp×Ph | <0.001 |
| Propulsive impulse (Ns/kg) | $A_{1st}$ | 0.85 ± 0.05 * | 0.80 ± 0.05 * | 0.77 ± 0.05 * | 0.75 ± 0.04 * | Sp>SOC, H>L | Gp | <0.001 |
| | $A_{2nd}$ | 0.29 ± 0.02 # | 0.27 ± 0.03 # | 0.27 ± 0.03 # | 0.29 ± 0.03 # | | Ph | <0.001 |
| | MS | 0.23 ± 0.02 | 0.22 ± 0.03 | 0.22 ± 0.02 | 0.23 ± 0.02 | | Gp×Ph | <0.001 |
| Stance-averaged antero-posterior force (N/kg) | $A_{1st}$ | 5.06 ± 0.48 * | 5.11 ± 0.32 * | 4.60 ± 0.39 * | 4.23 ± 0.39 * | Sp,H>M>L | Gp | <0.001 |
| | $A_{2nd}$ | 1.56 ± 0.24 # | 1.48 ± 0.32 # | 1.32 ± 0.40 # | 1.29 ± 0.37 # | | Ph | <0.001 |
| | MS | 0.31 ± 0.13 | 0.20 ± 0.32 | 0.07 ± 0.26 | 0.18 ± 0.25 | | Gp×Ph | <0.001 |
| Braking force (N/kg) | $A_{1st}$ | -1.46 ± 0.81 * | -1.63 ± 0.73 * | -1.53 ± 0.74 * | -1.96 ± 0.69 * | | Gp | 0.003 |
| | $A_{2nd}$ | -3.31 ± 0.86 # | -2.41 ± 0.29 # | -2.34 ± 0.47 # | -2.58 ± 0.77 # | Sp>Soc | Ph | <0.001 |
| | MS | -4.48 ± 0.54 | -3.74 ± 0.40 | -3.70 ± 0.41 | -3.60 ± 0.39 | Sp>Soc | Gp×Ph | <0.001 |
| Propulsive force (N/kg) | $A_{1st}$ | 6.05 ± 0.55 * | 6.10 ± 0.31 * | 5.48 ± 0.43 * | 5.06 ± 0.45 * | Sp,H>M>L | Gp | <0.001 |
| | $A_{2nd}$ | 4.81 ± 0.45 # | 4.62 ± 0.33 # | 4.37 ± 0.30 # | 4.24 ± 0.31 # | Sp>M,L, H>L | Ph | <0.001 |
| | MS | 4.59 ± 0.43 | 4.16 ± 0.44 | 3.95 ± 0.36 | 3.92 ± 0.36 | Sp>Soc | Gp×Ph | <0.001 |
| Vertical impulse (Ns/kg) | $A_{1st}$ | 2.15 ± 0.12 * | 2.02 ± 0.14 | 2.13 ± 0.14 | 2.14 ± 0.13 $ | Sp,M,L>H | Gp | 0.001 |
| | $A_{2nd}$ | 2.02 ± 0.08 | 1.99 ± 0.10 | 2.06 ± 0.08 # | 2.11 ± 0.11 # | L>H | Ph | <0.001 |
| | MS | 2.06 ± 0.09 | 2.05 ± 0.11 | 2.15 ± 0.09 | 2.18 ± 0.10 | M,L>Sp,H | Gp×Ph | <0.001 |
| Vertical impulse during the braking phase (Ns/kg) | $A_{1st}$ | 0.10 ± 0.03 * | 0.10 ± 0.02 * | 0.11 ± 0.06 * | 0.10 ± 0.02 * | | Gp | <0.001 |
| | $A_{2nd}$ | 0.82 ± 0.18 # | 0.92 ± 0.12 # | 0.99 ± 0.10 # | 0.97 ± 0.21 # | M,L>Sp | Ph | <0.001 |
| | MS | 1.09 ± 0.09 | 1.17 ± 0.10 | 1.26 ± 0.07 | 1.24 ± 0.12 | M,L>Sp | Gp×Ph | <0.001 |
| Vertical impulse during the propulsive phase (Ns/kg) | $A_{1st}$ | 2.04 ± 0.13 * | 1.92 ± 0.15 * | 2.01 ± 0.14 * | 2.03 ± 0.14 * | | Gp | <0.001 |
| | $A_{2nd}$ | 1.17 ± 0.20 # | 1.03 ± 0.12 # | 1.04 ± 0.13 # | 1.11 ± 0.18 # | | Ph | <0.001 |
| | MS | 0.94 ± 0.09 | 0.85 ± 0.07 | 0.85 ± 0.08 | 0.91 ± 0.08 | | Gp×Ph | <0.001 |
| Stance-averaged vertical force (N/kg) | $A_{1st}$ | 13.6 ± 1.1 * | 13.7 ± 0.8 * | 13.4 ± 0.9 * | 13.0 ± 0.9 * | | Gp | 0.016 |
| | $A_{2nd}$ | 19.5 ± 1.4 # | 18.6 ± 0.9 # | 18.1 ± 0.8 # | 17.6 ± 1.1 # | Sp>M,L, H>L | Ph | <0.001 |
| | MS | 21.2 ± 1.5 | 19.4 ± 1.0 | 19.2 ± 1.0 | 18.5 ± 0.8 | Sp>SOC, H>L | Gp×Ph | <0.001 |
| Averaged vertical force during the braking phase (N/kg) | $A_{1st}$ | 5.3 ± 1.0 * | 5.6 ± 0.9 * | 5.5 ± 1.4 * | 5.7 ± 0.9 * | | Gp | <0.001 |
| | $A_{2nd}$ | 20.2 ± 1.5 # | 19.9 ± 1.7 # | 19.9 ± 1.3 # | 18.9 ± 2.3 # | | Ph | <0.001 |
| | MS | 25.0 ± 1.4 | 23.1 ± 1.5 | 23.2 ± 1.2 | 22.0 ± 1.2 | SP>M,L | Gp×Ph | <0.001 |
| Averaged vertical force during the propulsive phase (N/kg) | $A_{1st}$ | 16.3 ± 1.4 * | 16.4 ± 0.8 * | 15.8 ± 1.1 * | 15.1 ± 1.0 * | | Gp | <0.001 |
| | $A_{2nd}$ | 20.2 ± 2.2 # | 18.8 ± 1.4 # | 17.8 ± 1.2 # | 17.5 ± 1.5 # | Sp>Soc, H>L | Ph | <0.001 |
| | MS | 19.6 ± 2.0 | 17.3 ± 1.4 | 16.6 ± 1.3 | 16.4 ± 1.3 | Sp>Soc | Gp×Ph | <0.001 |

Values are means and SDs. $A_{1st}$, the analytical steps across the 1st to 4th steps;

$A_{2nd}$, the analytical steps across 4 steps corresponding to half the number of steps required to reach maximal speed

Gp, group; Ph, phase; Gp×Ph, group by phase interaction; Sp, sprinters; Soc, soccer players; H, the high-speed soccer players; M, the medium-speed soccer players; L, the low-speed soccer players

*, significant difference compared to the other phases; $, significant difference compared to the $A_{2nd}$; #, significant difference compared to the $A_{2nd}$; MS, the analytical steps across 4 steps to reach maximal speed

There were no significant differences between $SOC_{High}$ and $Sp$ and between $SOC_{High}$ and $SOC_{Low}$ in each of the net antero-posterior impulse, braking impulse and propulsive impulse (Table 2). $SOC_{High}$ had a lower braking force compared with $Sp$. Propulsive force was greater in $SOC_{High}$ compared to $SOC_{Low}$.

No significant differences were found between $SOC_{High}$ and $Sp$ in vertical impulse, vertical impulse during the braking or propulsive phases (Table 2). The averaged vertical force during the propulsive phase was lower in $SOC_{High}$ than in $Sp$. $SOC_{High}$ exhibited a lower vertical impulse and greater averaged vertical force and vertical force during the propulsive phase compared to $SOC_{Low}$.

### Maximal speed phase (*MS*)

Maximal running speed was lower in $SOC_{High}$ than in $Sp$ but higher than in $SOC_{Low}$ (Table 1). There was no significant group difference in the $\%MS_0$. $SOC_{High}$ exhibited a longer braking time compared to $Sp$, resulting in a longer ground contact time. Flight time was shorter in $SOC_{High}$ than in $Sp$. $SOC_{High}$ demonstrated a shorter step length compared to $Sp$, while no significant difference in step frequency was observed. Compared to $SOC_{Low}$, $SOC_{High}$ exhibited a shorter ground contact time. Braking time tended to be shorter in $SOC_{High}$ compared to $SOC_{Low}$ ($p = 0.058$). No significant difference was observed in flight time between both groups. $SOC_{High}$ exhibited a longer step length and a higher step frequency compared to $SOC_{Low}$.

There were no significant differences between $SOC_{High}$ and $Sp$, nor between $SOC_{High}$ and $SOC_{Low}$ in net antero-posterior impulse, braking impulse or propulsive impulse. Both braking and propulsive forces were lower in $SOC_{High}$ compared to $Sp$. There were no significant differences between $SOC_{High}$ and $SOC_{Low}$ in stance-averaged antero-posterior force, braking or propulsive forces.

There were no significant differences between $SOC_{High}$ and $Sp$ in vertical impulse, vertical impulses during the braking and propulsive phases. Stance-averaged vertical force was lower in $SOC_{High}$ than in $Sp$, primarily due to lower averaged vertical forces during the propulsive phases. The averaged vertical force during the braking phase tended to be lower in $SOC_{High}$ than in $Sp$ ($p = 0.060$). Vertical impulse was lower in $SOC_{High}$ than in $SOC_{Low}$. $SOC_{High}$ exhibited a greater averaged vertical force compared to $SOC_{Low}$.

### Comparison between fast-running soccer players and sprinters at the same running speed (Table 3)

Regardless of running speed, $\%MS_0$ was higher in $SOC_{High}$ compared to $Sp$. No significant differences in ground contact time and flight time were observed between the two groups at any running speed. At 9.0 m/s, braking time was longer but propulsive time tended to be shorter ($p = 0.062$) in $SOC_{High}$ than in $Sp$. There were no significant differences in step frequency or step length between $SOC_{High}$ and $Sp$ at any running speed, except for 7.5 m/s.

At 9.0 m/s, net antero-posterior impulse, stance-averaged antero-posterior and braking forces were lower in $SOC_{High}$ compared to $Sp$, but no significant differences were found between the two groups in those at other running speeds. Propulsive impulses at 8.0 m/s and more were lower, and propulsive forces at 7.5 m/s and over were lower in $SOC_{High}$ than in $Sp$. No significant differences were observed between the two groups in braking impulse at any running speed.

Compared to $Sp$, $SOC_{High}$ exhibited larger vertical impulse during the braking phase but lower vertical impulse during the propulsive phase at 9.0 m/s, while the corresponding difference was found in vertical impulse on ground contact. There were no significant differences between the two groups in vertical impulse during the braking or propulsive phases at any running speed. At 9.0 m/s, the averaged vertical force during the propulsive phase was lower in $SOC_{High}$ compared to $Sp$, while no significant differences were found between the two groups in the stance-averaged vertical and vertical forces during the braking phase.

### Comparison between fast- and slow-running soccer players at the same running speed (Table 4)

Regardless of running speed, $\%MS_0$ was lower in $SOC_{High}$ compared to $SOC_{Low}$. No significant differences were observed between the two groups in ground contact time and flight time. Braking time was shorter in $SOC_{High}$ compared to $SOC_{Low}$,

**Table 3. Comparison of independent variables at the same running speed between sprinters and high-speed soccer players.**

| | 7.5 m/s Sprinters | 7.5 m/s High-speed soccer players | | 8.0 m/s Sprinters | 8.0 m/s High-speed soccer players | sig | 8.5 m/s Sprinters | 8.5 m/s High-speed soccer players | sig | 9 m/s Sprinters | 9 m/s High-speed soccer players | sig |
|---|---|---|---|---|---|---|---|---|---|---|---|---|
| Distance from start line, m | 8 ± 2 | 7 ± 1 | | 10 ± 2 | 10 ± 1 | | 14 ± 3 | 14 ± 2 | | 21 ± 7 | 23 ± 6 | * |
| Running speed, m/s | 7.56 ± 0.03 | 7.55 ± 0.03 | | 8.05 ± 0.03 | 8.05 ± 0.03 | | 8.54 ± 0.02 | 8.55 ± 0.03 | | 9.06 ± 0.03 | 9.05 ± 0.02 | |
| $\%MS_0$, % | 72 ± 3 | **76 ± 3** | | 77 ± 4 | **81 ± 4** | * | 82 ± 4 | **86 ± 4** | * | 87 ± 4 | **91 ± 4** | * |
| Ground contact time, s | 0.122 ± 0.008 | 0.119 ± 0.007 | | 0.114 ± 0.007 | 0.114 ± 0.006 | | 0.108 ± 0.007 | 0.109 ± 0.006 | | 0.102 ± 0.006 | 0.104 ± 0.005 | |
| Braking time, s | 0.032 ± 0.009 | 0.026 ± 0.007 | | 0.034 ± 0.009 | 0.034 ± 0.010 | | 0.037 ± 0.010 | 0.041 ± 0.007 | | 0.039 ± 0.008 | **0.048 ± 0.004** | * |
| Propulsive time, s | 0.091 ± 0.008 | 0.094 ± 0.006 | | 0.082 ± 0.008 | 0.080 ± 0.010 | | 0.072 ± 0.008 | 0.069 ± 0.009 | | 0.063 ± 0.007 | 0.057 ± 0.005 | |
| Flight time, s | 0.097 ± 0.009 | 0.089 ± 0.010 | | 0.101 ± 0.008 | 0.094 ± 0.008 | | 0.105 ± 0.006 | 0.097 ± 0.008 | | 0.110 ± 0.006 | 0.104 ± 0.007 | |
| Step frequency, steps/s | 4.57 ± 0.21 | **4.81 ± 0.20** | | 4.65 ± 0.21 | 4.84 ± 0.20 | * | 4.70 ± 0.19 | 4.86 ± 0.18 | | 4.73 ± 0.20 | 4.83 ± 0.17 | |
| Step length, m | 1.66 ± 0.08 | **1.57 ± 0.07** | | 1.73 ± 0.08 | 1.67 ± 0.07 | * | 1.82 ± 0.08 | 1.76 ± 0.07 | | 1.92 ± 0.08 | 1.87 ± 0.07 | |
| Net antero-posterior impulse, Ns/kg | 0.42 ± 0.09 | 0.41 ± 0.04 | | 0.36 ± 0.07 | 0.32 ± 0.04 | | 0.28 ± 0.06 | 0.25 ± 0.04 | | 0.20 ± 0.06 | **0.15 ± 0.05** | * |
| Braking impulse, Ns/kg | -0.06 ± 0.02 | -0.05 ± 0.01 | | -0.07 ± 0.02 | -0.06 ± 0.02 | | -0.09 ± 0.02 | -0.08 ± 0.01 | | -0.12 ± 0.03 | -0.12 ± 0.02 | |
| Propulsive impulse, Ns/kg | 0.48 ± 0.08 | 0.46 ± 0.05 | | 0.42 ± 0.06 | **0.38 ± 0.05** | * | 0.37 ± 0.05 | **0.32 ± 0.05** | * | 0.32 ± 0.04 | **0.27 ± 0.04** | * |
| Stance-averaged antero-posterior force, N/kg | 3.45 ± 0.81 | 3.44 ± 0.42 | | 3.10 ± 0.62 | 2.82 ± 0.39 | | 2.62 ± 0.65 | 2.26 ± 0.35 | | 1.99 ± 0.65 | **1.43 ± 0.49** | * |
| Braking force, N/kg | -1.97 ± 0.63 | -1.97 ± 0.66 | | -2.21 ± 0.83 | -1.93 ± 0.80 | | -2.47 ± 0.71 | -1.93 ± 0.39 | * | -3.14 ± 0.82 | **-2.52 ± 0.38** | * |
| Propulsive force, N/kg | 5.24 ± 0.63 | **4.90 ± 0.43** | | 5.21 ± 0.45 | **4.76 ± 0.40** | | 5.15 ± 0.41 | **4.73 ± 0.28** | * | 5.10 ± 0.43 | **4.75 ± 0.37** | * |
| Vertical impulse, Ns/kg | 2.07 ± 0.10 | 1.98 ± 0.10 | | 2.04 ± 0.09 | 1.97 ± 0.10 | | 2.01 ± 0.10 | 1.96 ± 0.09 | | 2.02 ± 0.09 | 1.98 ± 0.09 | |

*(Continued)*

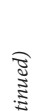

**Table 3.** (Continued)

| | 7.5 m/s | | 8.0 m/s | | 8.5 m/s | | 9 m/s | |
|---|---|---|---|---|---|---|---|---|
| | Sprinters | High-speed soccer players | Sprinters | High-speed soccer players | Sprinters | High-speed soccer players | Sprinters | High-speed soccer players |
| Vertical impulse during the braking phase, Ns/kg | 0.41 ± 0.21 | 0.25 ± 0.14 | 0.49 ± 0.22 | 0.46 ± 0.24 | 0.61 ± 0.26 | 0.69 ± 0.18 | 0.75 ± 0.25 | 0.95 ± 0.12 * |
| Vertical impulse during the propulsive phase, Ns/kg | 1.64 ± 0.22 | 1.70 ± 0.16 | 1.53 ± 0.24 | 1.48 ± 0.26 | 1.37 ± 0.26 | 1.24 ± 0.20 | 1.24 ± 0.25 | 1.00 ± 0.13 * |
| Vertical force, N/kg | 16.9 ± 1.3 | 16.5 ± 0.9 | 17.8 ± 1.3 | 17.2 ± 0.7 | 18.5 ± 1.2 | 17.9 ± 1.0 | 19.8 ± 1.3 | 18.9 ± 0.8 |
| Vertical force during the braking phase, N/kg | 12.1 ± 3.2 | **9.3 ± 1.9** | * 14.0 ± 3.2 | 12.9 ± 2.9 | 16.4 ± 3.4 | 16.6 ± 2.6 | 19.5 ± 3.0 | 20.4 ± 1.5 |
| Vertical force during the propulsive phase, N/kg | 19.3 ± 1.8 | 19.3 ± 1.3 | 20.1 ± 1.8 | 19.5 ± 1.4 | 20.4 ± 2.1 | 19.3 ± 1.5 | 20.9 ± 2.0 | 18.9 ± 1.6 * |

Values are means and SDs. * indicates a significant difference from sprinters.

while propulsive time was longer. $SOC_{High}$ exhibited higher step frequency compared to $SOC_{Low}$. Step length at 8.0 m/s tended to be shorter in $SOC_{High}$ than in $SOC_{Low}$, but the difference was not significant ($p = 0.070$).

At 7.5 m/s and 8.0 m/s, net antero-posterior impulse was greater in $SOC_{High}$ compared to $SOC_{Low}$, primarily due to lower braking impulse and larger propulsive impulse in the former. The stance-averaged antero-posterior forces at the two running speeds were higher in $SOC_{High}$ than in $SOC_{Low}$. At 8.0 m/s, propulsive force was higher in $SOC_{High}$ than in $SOC_{Low}$, and braking force tended to be lower in the former compared to the latter ($p = 0.060$).

At 7.5 m/s and 8.0 m/s, vertical impulses during the braking phase were lower and those during the propulsive phase were higher in $SOC_{High}$ than in $SOC_{Low}$. There was no significant difference in stance-averaged vertical force between the two groups. However, the averaged vertical force during the braking phase was smaller and that during propulsive phase was greater in $SOC_{High}$ compared to $SOC_{Low}$.

## Discussion

This study conducted group comparisons through the two approaches: (1) aligning sprint running phase based on definitions from a previous study [8] and (2) matching running speeds between $SOC_{High}$ and $Sp$ (7.5–9.0 m/s), as well as among soccer players (7.5 m/s and 9.0 m/s). The first approach revealed that running speed for $SOC_{High}$ was similar to that for $Sp$ during the $A_{1st}$ but was lower than $Sp$ during the $A_{2nd}$ and the $MS$ phases. The second approach indicated that at 9.0 m/s, $SOC_{High}$ exhibited smaller net antero-posterior impulse and propulsive impulse compared to $Sp$. In $SOC_{High}$, vertical impulse during the braking phase increased due to a longer braking time, whereas vertical impulse during the propulsive phase decreased due to both a shorter propulsive time and a lower vertical force during the corresponding phase. In the running phase from $A_{1st}$ to $MS$, running speeds were higher in $SOC_{High}$ than in $SOC_{Low}$. This was due to a higher step frequency throughout all sprint phases and a longer step length from the $A_{2nd}$ to the $MS$ in $SOC_{High}$. Additionally, $SOC_{High}$ exhibited a greater net antero-posterior impulse during the $A_{1st}$ and lower vertical impulse throughout entire running phase compared to $SOC_{Low}$. These results were consistent regardless of the approaches. Thus, the current results indicate that as compared to sprinters, fast-running soccer players may exhibit different sprint mechanics for speed acquisition during the initial acceleration phase, although they can achieve similar running speeds and maintain comparable acceleration capability up to 8.5 m/s, but beyond the 2nd acceleration phase, they generate lower propulsive forces and greater vertical force production during the braking phase due to a longer braking time compared to sprinters.

### Initial sprint acceleration phase

In the $A_{1st}$, $SOC_{High}$ exhibited higher running speed compared with $SOC_{Low}$. Running speed is theoretically the product of step length and step frequency. Step frequency was higher in $SOC_{High}$ than in $SOC_{Low}$, whereas no significant difference was found in step length between the two groups. Therefore, group difference in running speed depends on step frequency. Consequently, a shorter ground contact time resulted in a higher step frequency for $SOC_{High}$ compared with $SOC_{Low}$. Additionally, in this study, the antero-posterior impulse was higher in $SOC_{High}$ than in $SOC_{Low}$. Step length is known to be associated with both antero-posterior and vertical impulses [22]. A lack of group difference in step length could be due to a lower vertical impulse in $SOC_{High}$ in comparison with $SOC_{Low}$. Considering that stance-averaged antero-posterior force was greater in $SOC_{High}$ than in $SOC_{Low}$, it is likely that $SOC_{High}$ can generate greater antero-posterior force than $SOC_{Low}$ even within a short duration of ground contact in the 1st acceleration phase.

The $SOC_{High}$ demonstrated a running speed comparable to that of $Sp$. The step frequency was higher in $SOC_{High}$ compared with $Sp$, resulting from a shorter ground contact time. On the other hand, $SOC_{High}$ had a lower net antero-posterior impulse compared to $Sp$, primarily due to a lower propulsive impulse. There were no significant differences in stance-averaged antero-posterior force, baking or propulsive forces between the two groups. Therefore, fast-running soccer players may be able to achieve similar speeds to sprinters by reducing ground contact time, particularly during the propulsive phase, and increasing step frequency.

Table 4. Comparison of independent variables at the same running speed between low- or medium- and high-speed soccer players.

| | 7.5 m/s | | | | 8.0 m/s | | | |
|---|---|---|---|---|---|---|---|---|
| | Low-speed soccer players | Medium-speed soccer players | High-speed soccer players | | Low-speed soccer players | Medium-speed soccer players | High-speed soccer players | |
| Distance from start line, m | 12 ± 2 | 9 ± 1 | 7 ± 1 | * | 21 ± 8 | 14 ± 2 | 10 ± 1 | $ |
| Running speed, m/s | 7.54 ± 0.03 | 7.54 ± 0.02 | 7.55 ± 0.03 | | 8.05 ± 0.03 | 8.05 ± 0.03 | 8.05 ± 0.03 | |
| %MS$_0$, % | 85 ± 3 | 81 ± 3 | 76 ± 3 | $ | 90 ± 3 | 87 ± 3 | 81 ± 4 | $ |
| Ground contact time, s | 0.125 ± 0.007 | 0.125 ± 0.007 | 0.119 ± 0.007 | | 0.118 ± 0.006 | 0.118 ± 0.007 | 0.114 ± 0.006 | |
| Braking time, s | 0.048 ± 0.010 | 0.043 ± 0.009 | 0.026 ± 0.007 | $ | 0.052 ± 0.006 | 0.050 ± 0.006 | 0.034 ± 0.010 | $ |
| Propulsive time, s | 0.078 ± 0.008 | 0.083 ± 0.010 | 0.094 ± 0.006 | $ | 0.067 ± 0.006 | 0.069 ± 0.006 | 0.080 ± 0.010 | $ |
| Flight time, s | 0.093 ± 0.009 | 0.095 ± 0.007 | 0.089 ± 0.010 | | 0.098 ± 0.008 | 0.100 ± 0.008 | 0.094 ± 0.008 | |
| Step frequency, steps/s | 4.61 ± 0.18 | 4.57 ± 0.18 | 4.81 ± 0.20 | $ | 4.64 ± 0.17 | 4.58 ± 0.16 | 4.84 ± 0.20 | $ |
| Step length, m | 1.64 ± 0.06 | 1.65 ± 0.06 | 1.57 ± 0.07 | # | 1.74 ± 0.06 | 1.76 ± 0.06 | 1.67 ± 0.07 | |
| Net antero-posterior impulse, Ns/kg | 0.24 ± 0.05 | 0.32 ± 0.05 | 0.41 ± 0.04 | $ | 0.15 ± 0.06 | 0.22 ± 0.04 | 0.32 ± 0.04 | $ |
| Braking impulse, Ns/kg | -0.09 ± 0.03 | -0.05 ± 0.02 | -0.05 ± 0.01 | | -0.13 ± 0.04 | -0.09 ± 0.02 | -0.06 ± 0.02 | $ |
| Propulsive impulse, Ns/kg | 0.33 ± 0.03 | 0.37 ± 0.05 | 0.46 ± 0.05 | $ | 0.29 ± 0.03 | 0.31 ± 0.03 | 0.38 ± 0.05 | $ |
| Stance-averaged antero-posterior force, N/kg | 1.90 ± 0.44 | 2.54 ± 0.35 | 3.44 ± 0.42 | $ | 1.30 ± 0.46 | 1.85 ± 0.36 | 2.82 ± 0.39 | $ |
| Braking force, N/kg | -1.98 ± 0.80 | -1.43 ± 0.81 | -1.97 ± 0.66 | | -2.54 ± 0.66 | -1.79 ± 0.44 | -1.93 ± 0.80 | |
| Propulsive force, N/kg | 4.25 ± 0.37 | 4.45 ± 0.31 | 4.90 ± 0.43 | $ | 4.27 ± 0.38 | 4.45 ± 0.31 | 4.76 ± 0.40 | * |
| Vertical impulse, Ns/kg | 2.08 ± 0.10 | 2.08 ± 0.10 | 1.98 ± 0.10 | $ | 2.08 ± 0.09 | 2.09 ± 0.10 | 1.97 ± 0.10 | $ |
| Vertical impulse during the braking phase, Ns/kg | 0.79 ± 0.25 | 0.66 ± 0.21 | 0.25 ± 0.14 | $ | 0.97 ± 0.16 | 0.91 ± 0.13 | 0.46 ± 0.24 | $ |
| Vertical impulse during the propulsive phase, Ns/kg | 1.26 ± 0.20 | 1.40 ± 0.21 | 1.70 ± 0.16 | $ | 1.09 ± 0.14 | 1.15 ± 0.13 | 1.48 ± 0.26 | $ |
| Vertical force, N/kg | 16.5 ± 1.0 | 16.5 ± 0.8 | 16.5 ± 0.9 | | 17.4 ± 0.9 | 17.5 ± 1.0 | 17.2 ± 0.7 | |
| Vertical force during the braking phase, N/kg | 16.3 ± 2.8 | 15.0 ± 2.9 | 9.3 ± 1.9 | $ | 19.0 ± 1.9 | 18.5 ± 1.8 | 12.9 ± 2.9 | $ |
| Vertical force during the propulsive phase, N/kg | 17.3 ± 1.5 | 17.8 ± 1.0 | 19.3 ± 1.3 | $ | 17.3 ± 1.2 | 17.8 ± 1.1 | 19.5 ± 1.4 | $ |

Values are means and SDs. * indicates a significant difference from low-speed soccer players. # indicates a significant difference from medium-speed soccer players. $ indicates significant differences from both low- and medium-speed soccer players.

**The 2nd acceleration phase**

During the $A_{2nd}$, $SOC_{High}$ exhibited a higher running speed compared to $SOC_{Low}$. The mechanism contributing to the difference in running speed between the two groups was associated with the magnitude of step frequency and step length. Compared to $SOC_{Low}$, $SOC_{High}$ demonstrated higher step frequency and longer step length. $SOC_{High}$ had shorter ground contact time, and higher stance-averaged vertical force as compared to $SOC_{Low}$. The stance-averaged vertical force was negatively related to ground contact time in the $A_{2nd}$ ($r = 0.726$ for soccer players) (unpresented data). These indicate that as compared to $SOC_{Low}$, $SOC_{High}$ can produce greater vertical force within a short ground contact time. Additionally, since there was no significant difference in the flight time between $SOC_{High}$ and $SOC_{Low}$, a higher step frequency for $SOC_{High}$ may be due to a shorter ground contact time than $SOC_{Low}$. The vertical and antero-posterior impulses could not explain the group differences in step length. Net antero-posterior impulse is associated with the subsequent change in running speed during the entire acceleration phase [23]. The lack of difference in antero-posterior impulse indicates that there was no difference in the change in running speed. Additionally, the stance-averaged antero-posterior force in the first step correlates with each 5-m split time from 5 m to 20 m for semi-professional soccer players [24]. Thus, the difference in the antero-posterior impulse before the 2nd acceleration phase might contribute to the observed difference in running speed. In addition, antero-posterior impulse is a determinant of step length [22]. In this study, the net antero-posterior impulse in the initial acceleration phase was higher in $SOC_{High}$ compared with $SOC_{Low}$. It implies that group difference in step length might be due to the antero-posterior impulse before the 2nd acceleration phase. Therefore, a higher running speed observed in $SOC_{High}$ during the 2nd acceleration phase could be due to a longer step length, driven by a greater antero-posterior impulse before the $A_{2nd}$, as well as a higher step frequency, resulting from shorter ground contact time and greater vertical force production.

$SOC_{High}$ exhibited a lower running speed compared to $Sp$. Step length was lower in $SOC_{High}$ than in $Sp$, whereas no significant difference was found in step frequency between both groups. Therefore, the difference in the running speed between $SOC_{High}$ and $Sp$ was due to differences in the step length. In the 1st acceleration phase, $SOC_{High}$ had a higher step frequency compared to $Sp$, but there was no significant difference in step frequency between the two groups in the 2nd acceleration phase. This could be due to the lack of group differences in ground contact time and flight time during the 2nd acceleration phase. On the other hand, while no significant differences in antero-posterior and vertical impulses were observed, $SOC_{High}$ had a lower step length compared to $Sp$. As for the antero-posterior components, a previous study demonstrated that soccer players exhibit lower net antero-posterior impulse than $Sp$, and the force vector is more inclined in the direction of propulsion in high-speed running (8 and 8.5 m/s) [15]. This discrepancy may stem from differences in analytical approaches: comparing at the same running speed vs. within the same sprint phase. To examine this assumption, we plotted the relationship between each measured variable and $\%MS_0$ (S1 Fig). The results showed that both net antero-posterior impulse and stance-averaged antero-posterior force exhibited linear relationships, regardless of the group. This suggests that, despite differences in absolute running speed, the ability to accelerate within the same running phase may be comparable between sprinters and fast-running soccer players. As mentioned above, the group differences in step length may have been attributable to the antero-posterior impulse before the 2nd acceleration phase. Therefore, the difference in running speed between $SOC_{High}$ and $Sp$ during the 2nd acceleration phase could be due to a greater antero-posterior impulse before the $A_{2nd}$.

**Maximal speed phase**

In the $MS$, $SOC_{High}$ demonstrated a higher running speed compared to $SOC_{Low}$, resulting from a longer step length and a higher step frequency. The higher step frequency observed in $SOC_{High}$ can be attributed to a shorter ground contact time. It is known that footballers with high vertical stiffness during hopping tasks have higher maximal speed compared to footballers with low vertical stiffness, and the group differences in sprint performance can be observed at a distance of 30–60 m [25]. Combining the current results with the previous findings, it can be suggested that $SOC_{High}$ might have higher vertical

stiffness via stance-averaged vertical forces compared with $SOC_{Low}$. Additionally, since no significant group differences in antero-posterior and vertical impulses were found between the two groups, longer step length in $SOC_{High}$ might result from the magnitude of running speed before maximal speed phase. Therefore, $SOC_{High}$ can produce greater vertical force during a short ground contact compared with $SOC_{Low}$ in the maximal speed phase.

The difference in running speed between $SOC_{High}$ and $Sp$ mirrored that in the $A_{2nd}$. Step length was shorter in $SOC_{High}$ than in $Sp$, whereas no significant difference was found in step frequency between both groups. These findings support the previous findings that during the maximal speed phase, non-sprinters demonstrated shorter step lengths whereas step frequency of non-sprinters was comparable to that of sprinters [26]. The difference in running speed was due to the magnitude of step length between the two groups. Shorter step length for $SOC_{High}$ in comparison with $Sp$ might be due to the difference in running speed before the maximal speed phase. $SOC_{High}$ exhibited vertical impulse comparable to $Sp$, but $SOC_{High}$ had a longer ground contact time and a shorter flight time. Since step frequency is the inverse of the sum of ground contact time and flight time, the longer ground contact and shorter flight time offset each other, resulting in no difference between $SOC_{High}$ and $Sp$ for step frequency. A previous study found that trained team sport athletes exhibited lower vertical stiffness compared with sprinters [26]. A stiffer leg spring allows high vertical forces [27] and a short contact time [28]. In this study, the stance-averaged vertical force was lower in $SOC_{High}$ compared to $Sp$, which is consistent with the previous findings [16,29]. The vertical force during the braking phase tended to be lower in $SOC_{High}$ than in $Sp$. Clark and Weyand [16] demonstrated a positive correlation between maximal running speed and stance-averaged vertical force during the braking phase. Additionally, $SOC_{High}$ exhibited a lower vertical force during the propulsive phase compared to $Sp$. The development of maximal speed during sprinting is accompanied by an increase in vertical stiffness [30]. Based on these findings, it is likely that fast-running soccer players cannot stiffen their lower limbs particularly during the maximal speed phase, and consequently generate lower vertical forces, and experience longer ground contact times compared to sprinters.

### Factors contributing to differences in maximal running speed between fast-running soccer players and sprinters

In this study, we compared spatiotemporal variables and ground reaction forces between $SOC_{High}$ and $Sp$ with relation to the same running speed and the same running phase. In this section, we discuss the factors contributing to the lower maximal running speed of $SOC_{High}$ compared to $Sp$, based on the findings of this study. As seen in Fig 1, $SOC_{High}$ exhibited a higher step frequency than $Sp$ up to 7.5 m/s; however, beyond this speed, no significant difference was observed. Additionally, in $SOC_{High}$, step frequency decreased from $A_{2nd}$ to $MS$, whereas it remained constant during the corresponding phase in $Sp$. This decrease may be attributable to a longer ground contact time in $SOC_{High}$, likely due to lower vertical forces, particularly during the propulsive phase. As greater vertical force during ground contact was associated with a shorter ground contact time above mentioned, it is assumed that $SOC_{High}$ may be unable to reduce their ground contact time due to lower vertical force during the propulsive phase, which may have prevented them from maintaining step frequency.

Another mechanism underlying the difference in maximal running speed between $SOC_{High}$ and $Sp$ may be the difference in net antero-posterior impulse at the same running speed. At 9.0 m/s, $SOC_{High}$ exhibited a lower net antero-posterior impulse, primarily due to a lower propulsive impulse compared to $Sp$. Given these findings, the lower maximal running speed in $SOC_{High}$ may be attributable to the inability to generate sufficient acceleration to achieve higher speeds beyond $A_{2nd}$. Taken together, fast-running soccer players exhibit lower acceleration capability beyond the 2nd acceleration phase and reduced vertical force during the propulsive phase, which may contribute to their lower maximal running speed compared to sprinters.

### Factors contributing to differences in maximal running speed between fast- and slow-running soccer players

In this section, we discuss the factors contributing to the lower maximal running speed of $SOC_{High}$ compared to $SOC_{Low}$, based on the findings of this study. As shown in Fig 1, $SOC_{High}$ exhibited a higher step frequency during entire sprint

running. This may be due to a shorter flight time in $SOC_{High}$, primarily resulting from lower vertical impulse, as no significant differences were observed in ground contact time between the two groups. Additionally, beyond $A_{2nd}$, $SOC_{High}$ exhibited higher vertical forces during ground contact, particularly in the propulsive phase, leading to a shorter ground contact time compared to $SOC_{Low}$. Therefore, the higher step frequency in $SOC_{High}$ beyond the 2nd acceleration phase can be attributed to greater vertical force.

In both groups, step length increased with running speed, and $SOC_{High}$ was able to extend their step length even further as running speed increased. Both antero-posterior and vertical impulses are determinants of step length [22]. Since vertical impulse was lower in $SOC_{High}$ compared to $SOC_{Low}$, the longer step length in $SOC_{High}$ may be attributed to a greater net antero-posterior impulse. In particular, $SOC_{High}$ exhibited less braking impulse due to brief braking time and greater propulsive impulse resulting from both longer propulsive time and propulsive force up to the $A_{2nd}$. Based on these findings, fast-running soccer players can produce greater propulsive force throughout the entire running phase and generate larger vertical forces at higher speeds compared to slow-running soccer players.

## Limitations of this study

There are some limitations in this study. First, we did not have data on sprint running kinematics. A study by Garden [31] has shown that the kinematics of sprint running may differ between sprinters and soccer players, which could explain the differences in spatiotemporal and mechanical variables observed in this study. In particular, biomechanical analysis is needed to investigate the factors contributing to the increase in braking time and the reduced propulsive time from the 2nd acceleration phase to the maximal speed phase. Second, the difference in ground reaction forces between sprinters and soccer players might be influenced by footwear (spiked shoes vs. running shoes). To the best of our knowledge, no study has explicitly examined this factor. In this study, participants were instructed to wear their regularly used shoes to ensure familiarity and natural sprint performance. Finally, this study examined regional collegiate soccer players, whereas the previous study included regional to national level players [17]. Further research is needed to clarify these points.

## Conclusion

Sprint mechanics of fast-running soccer players is characterized by similar ability of speed acquisition up to the 2nd acceleration to sprinters but, at above 9.0 m/s and more, they exhibit an increase in vertical impulse during ground contact, leading to a lower step frequency. Additionally, fast-running soccer players produce greater acceleration capacity, lower vertical force during the braking phase, and higher vertical force during the propulsive phase compared to slow-running soccer players.

## Supporting information

**S1 Fig. Associations of spatiotemporal variables and ground reaction forces with the percentage of running speed relative to theoretical maximal running speed.** Data points are represented as sprinters (white-filled triangles), fast-running soccer players (black-filled circles), medium-running soccer players (gray-filled circles), and slow-running soccer players (white-filled circles).
(DOCX)

**S1 Dataset. SuppInfo_PlosOne DataSet.**
(XLSX)

## Acknowledgements

This study was supported by a NIFS project for the assessment of physical fitness for athletes.

## Author contributions

**Conceptualization:** Yohei Takai, Norihide Sugisaki, Hiroaki Kanehisa.

**Data curation:** Takaya Yoshimoto, Naotoshi Mitsukawa, Kai Kobayashi, Hiroyasu Tsuchie.

**Formal analysis:** Terumitsu Miyazaki, Hiroyasu Tsuchie.

**Investigation:** Takaya Yoshimoto, Naotoshi Mitsukawa, Kai Kobayashi, Hiroyasu Tsuchie.

**Methodology:** Yohei Takai, Terumitsu Miyazaki, Norihide Sugisaki, Naotoshi Mitsukawa.

**Writing – original draft:** Yohei Takai.

**Writing – review & editing:** Norihide Sugisaki, Naotoshi Mitsukawa, Hiroaki Kanehisa.

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
