## [Decision Letter · Decision Letter 0]

20 Dec 2024

PONE-D-24-37320Spatiotemporal and kinetic characteristics during maximal sprint running in fast running soccer playersPLOS ONE

Dear Dr. Takai,

Thank you for submitting your manuscript to PLOS ONE. After careful consideration, we feel that it has merit but does not fully meet PLOS ONE’s publication criteria as it currently stands. Therefore, we invite you to submit a revised version of the manuscript that addresses the points raised during the review process.

We look forward to receiving your revised manuscript.

Kind regards,

Hasan Sozen

Academic Editor

PLOS ONE

Journal Requirements:

3. In the online submission form, you indicated that the data that support the findings of this study are available from the corresponding author upon reasonable request.

Reviewers' comments:

Reviewer's Responses to Questions

**Comments to the Author**

1. Is the manuscript technically sound, and do the data support the conclusions?

Reviewer #1: Yes

Reviewer #2: Yes

2. Has the statistical analysis been performed appropriately and rigorously? 

Reviewer #1: Yes

Reviewer #2: Yes

3. Have the authors made all data underlying the findings in their manuscript fully available?

Reviewer #1: No

Reviewer #2: Yes

4. Is the manuscript presented in an intelligible fashion and written in standard English?

Reviewer #1: Yes

Reviewer #2: Yes

5. Review Comments to the Author

Reviewer #1: General comments

This manuscript compared spatiotemporal and kinetic variables during sprint running in soccer players and sprint runners. The manuscript reports important data on sprint running and contains valuable information. However, there are specific questions and concerns that should be considered. I think the analyses of the 2nd acceleration phase are insignificant in this manuscript. Although %V0 in the 2nd acceleration phase was similar among the groups, the absolute running velocity was significantly different among the groups. Such differences in the running velocity made the group comparison difficult to interpret. I suggest comparing the groups on a phase of equal absolute velocity.

Specific comments

L37

The vertical ground force was “smaller” in SOChigh than in sprinters, not “reduced”.

L38-40

I couldn’t understand this conclusion. In the present results, the vertical ground force during the maximal speed phase was significantly smaller in SOChigh than in sprinters (Table 2).

L54

Although the authors defined three primary phases, the text after this point described only two phases (acceleration and maximal speed phases).

L68-72

Please provide the phases (the 1st acceleration, the 2nd acceleration or the maximal speed phases) of these findings.

L73-75

Please provide appropriate references for this statement.

L75-76

Please add appropriate reference(s) for this statement.

L96

Is “The participants” the sprinters?

L116-117

How do the authors think about the effects of different types of shoes on the present findings?

L133-134

How much did the center of pressure move in a single step? If the position of the center of pressure moved from the time of ground contact to the takeoff, it should affect the step length determined by the method described here.

P141-142

“speed” and “velocity” are mixed in this manuscript. The typical example is “maximal speed (Vmax)”. However, these terms are distinctly different. “speed” is a scalar and “velocity” is a vector.

L146-147

Is this correct? For example, is the slope F0?

L147

Please include appropriate reference(s) for the Pmax calculation.

L160

What is “time” in this context? Phase?

L168

“IBM”

Results

Showing the ground reaction force profiles during a single step for each phase may help the readers understand the results.

Figure 1

Please provide a higher-resolution image.

L208-209

What is “the stance-averaged antero-posterior force net antero-posterior impulse”?

L235-241

There was no discussion regarding the horizontal force-velocity profile. The authors should reconsider the need for this data.

L257-262

I think the group difference in step frequency was simply due to the difference in contact time. There is no need to discussion of vertical impulse.

L270-271

The effect size is only reported in limited places in the manuscript. Please provide the effect size for the other comparisons.

L272-276

Again, I think the group difference in step frequency was simply due to the difference in contact time.

L287-289

How about sprint running on the ground, not the treadmill?

L293-296

Please specify the phase of the sprint running regarding the previous findings.

L300-304

The impulse is equal to the change in momentum (mass times velocity). But the running velocity itself was different among the groups in the 2nd acceleration phase. Hence, it is difficult to discuss the group difference from the viewpoint of impulse (equal to changes in velocity). I suggest to compare the groups on a phase of equal absolute velocity (e.g., 7.5-7.9 m/s), not on a phase of equal relative velocity (%V0). At 7.9 m/s of running velocity, SOClow cannot accelerate at all, but SOChigh can still accelerate. The difference and its mechanism are scientifically interesting.

L304

“shorter” step length rather than “lower”

L318-321

Again, I suggest to directly (statistically) compare the groups on a phase of equal absolute velocity (e.g., 8.5-8.8 m/s), as the authors stated.

L347-349

Are these previous findings obtained from sprint running?

L354-356

Sprinters exhibited greater vertical force not only in the braking phase but also in the propulsive phase.

L356-358

I think that the vertical force and reactive strength are not independent with each other. The greater vertical force during sprint running may result from the greater reactive strength.

L359

What is “requisite reactive strength”?

L370-373

The lack of significant difference is NEVER the study limitation. The lack of discussion about the reported data is a serious flow of the manuscript.

L381-382

From which data the authors state so? “6 m/s”?

L382-384

The authors did not state so in the discussion. Not only SOChigh, but all the groups showed decreases in the step frequency.

L397

No significant difference was found in the step frequency during the 2nd acceleration phase.

Reviewer #2: Spatiotemporal and kinetic characteristics during maximal sprint running in fast running

soccer players.

The work has been well described and presented, although many of results were already known and for a main part not very surprising, the work comprises an interesting data set about differences in sprint execution of soccer players and sprint athletes. The soccer players are split into three groups based on maximal sprint speeds, consequently many differences would be anticipated among the groups. Which indeed was the case. The authors carefully interpret their results, recognizing the importance of differences in absolute speed and the dependencies of results found in the different sprint phases on the preceding sprint phase.

I found the comparison between fast soccer players and sprinters the most interesting, with the most interesting finding:’’Fast-running soccer players exhibited propulsion forces similar to that of sprinters during the initial acceleration phase, but their vertical ground forces were comparatively less in the maximal speed.”

A limitation is that the soccer players wore running shoes and the sprinters spikes, but I anticipate this will be discussed. Moreover, many of the more subtle differences found among the groups probably are dependent upon these specific groups of soccer players and sprinters.

1. I noticed that this point wasn’t addressed at all , I think it should be discussed.

2. Although the methodology has been used and described in previous work, I still think that the manuscript would be improved with inclusion of a figure showing ground reaction forces and some explanation of the different parameters investigated.

3. I do not think that table 3 adds very much (I am aware that this force profiling is frequently used but in a way it just summarizes differences in acceleration and maximal speed capacity among athletes. It involves extrapolation and the assumption of linearity. I am curious to read how the authors are going to explain differences and/ or lack thereof among groups. (lines 367-375), I am slightly disappointed, why not pointing out that this method has inherent uncertainties? Why not just leave Table 3 and these results out of the study, which would also give room for a figure (point 2)

Perspective:’ Therefore,conducting assisted sprinting and plyometric training that focuses on the vertical component of force development might benefit for fast-running soccer players in improving maximal running speed.’

4.I think the authors should address that maximal sprint speed is of far less importance for soccer players than acceleration, it may not be warranted to invest much specific training to increase Vmax for soccer players.

Minor

Typo error in the following, surely fast running soccer player produce vertical forces

in the 2nd acceleration and the maximal speed phases,fast-running soccer players would not produce vertical force and would have lower step frequency andstep length compared with sprinters.

6. PLOS authors have the option to publish the peer review history of their article (what does this mean? ). If published, this will include your full peer review and any attached files.

**Do you want your identity to be public for this peer review?** For information about this choice, including consent withdrawal, please see our Privacy Policy .

Reviewer #1: No

Reviewer #2: No

---

## [Author Response · Author response to Decision Letter 1]

4 Mar 2025

We appreciate your review of our manuscript. Please find attached the files containing the revised text and our responses to the reviewers' comments.

---

## [Editor Report · Decision Letter 1]

18 Mar 2025

Spatiotemporal and kinetic characteristics during maximal sprint running in fast running soccer players

PONE-D-24-37320R1

Dear Dr. Takai,

We’re pleased to inform you that your manuscript has been judged scientifically suitable for publication and will be formally accepted for publication once it meets all outstanding technical requirements.

Kind regards,

Hasan Sozen

Academic Editor

PLOS ONE

---

## [Editor Report · Acceptance letter]

PONE-D-24-37320R1

PLOS ONE

Dear Dr. Takai,

I'm pleased to inform you that your manuscript has been deemed suitable for publication in PLOS ONE. Congratulations! Your manuscript is now being handed over to our production team.

Kind regards,

on behalf of

Assoc. Prof. Hasan Sozen

Academic Editor

PLOS ONE
